# Surface Modification of γ-Al_2_O_3_ Nanoparticles Using Conductive Polyaniline Doped by Dodecylbenzene Sulfonic Acid

**DOI:** 10.3390/polym14112232

**Published:** 2022-05-31

**Authors:** Cheng-Ho Chen, Ying-Chen Lin, Hung-Mao Lin

**Affiliations:** Department of Chemical and Materials Engineering, Southern Taiwan University of Science and Technology, Tainan City 710, Taiwan; ma340101@stust.edu.tw (Y.-C.L.); hmlin@stust.edu.tw (H.-M.L.)

**Keywords:** polyaniline, dodecylbenzene sulfonic acid, γ-Al_2_O_3_, in situ polymerization, core–shell nanocomposite

## Abstract

In this study, electrically conductive PANDB/γ-Al_2_O_3_ core–shell nanocomposites were synthesized by surface modification of γ-Al_2_O_3_ nanoparticles using polyaniline doped with dodecylbenzene sulfonic acid. The PANDB/γ-Al_2_O_3_ core–shell nanocomposites were synthesized by in situ polymerization. Pure PANDB and the PANDB/γ-Al_2_O_3_ core–shell nanocomposites were characterized using Fourier transform infrared spectroscopy, ultraviolet–visible spectroscopy, transmission electron microscopy, field emission scanning electron microscopy, and measurement of a four-point probe. The conductivity of the PANDB/γ-Al_2_O_3_ core–shell nanocomposite was about 0.72 S/cm when the weight ratio of aniline/γ-Al_2_O_3_ was 3/1. The results showed that the conductivity of the PANDB/γ-Al_2_O_3_ core–shell nanocomposite decreased with increasing amounts of γ-Al_2_O_3_ nanoparticles. The transmission electron microscopy results indicated that the γ-Al_2_O_3_ nanoparticles were thoroughly coated with PANDB to form a core–shell structure. Transmission electron microscopy and field emission scanning electron microscopy images of the conductive PANDB/γ-Al_2_O_3_ core–shell nanocomposites also showed that the thickness of the PANDB layer decreased as the amount of γ-Al_2_O_3_ was increased.

## 1. Introduction

Most polymers are insulators because they are made of covalent bonds without free- moving electrons or ions. Inherently conductive polymers (ICPs) are a special class of synthetic polymers with unique electro-optical characteristics. ICPs possess conjugated chains with alternating single and double bonds [1]. Polyaniline (PANI) is an important member of the intrinsically conductive polymer (ICP) family. Since PANI is easy to synthesize and exhibits a wide range of conductivity, low operational voltage, unique electrochemical properties, and environmental stability, numerous researchers have studied it extensively [2,3,4,5,6] and have used it in many applications, such as secondary batteries [7,8], biosensors [9,10], corrosion protectors [11,12], antistatic packaging materials [13], and light-emitting diodes (LEDs) [14].

The encapsulation of inorganic materials inside a PANI shell has become the most popular and interesting aspect of nanocomposites in recent years. The combination of inorganic components with electrically conductive PANI (emeraldine salt (ES)-type PANI) has attracted considerable attention due to the novel physical and chemical properties of the resulting nanocomposite and potential applications. It has great potential applications in the fields of medication delivery, biosensors, chemical assembly, materials science, etc. These core–shell conductive nanocomposites can provide new synergistic properties that cannot be obtained from individual materials alone. Many groups have reported PANI/inorganic core–shell nanocomposites such as PANI/bagasse fiber (BF) [15], PANI/Y_2_O_3_ [16], PANI/NiCo_2_O_4_ [17], PANI/T-ZnOw [18], PANI/TiO_2_ [19], PANI/MnO_2_ [20], PANI/alumina [21,22,23,24], and PANI/clay [25,26].

However, PANI has some processing disadvantages, such as low or no solubility in most common organic solvents and poor processing properties [1]. Many researchers have attempted to synthesize soluble conductive PANI doped with various dopants. The most promising and attractive approach is to synthesize PANI doped with dodecylbenzene sulfonic acid (DBSA) (PANDB) in aqueous solution [27] or emulsion polymerization [28]. In these synthetic methods, DBSA can act as both a surfactant and a dopant during the synthesis. Since the molecular chain of DBSA has a lipophilic group (–C_12_H_25_), PANDB-based materials are soluble in common organic solvents. Therefore, PANDB-based materials can have more diverse applications.

Aluminum oxide (Al_2_O_3_) has good physical properties, such as thermal stability, corrosion resistance, abrasion resistance, electrical insulation, and high mechanical strength. Therefore, Al_2_O_3_ is the material most commonly applied in industrial applications. γ-Al_2_O_3_ is one of the metastable polymorphs of transition alumina, which has higher thermal stability compared with other polymorphs of Al_2_O_3_.

To our knowledge, no studies have been reported on the preparation of conductive PANDB/γ-Al_2_O_3_ core–shell nanocomposites via in situ polymerization of aniline (AN) in the presence of DBSA. In this study, γ-Al_2_O_3_ nanoparticles are embedded by PANDB via in situ polymerization to form conductive PANDB/γ-Al_2_O_3_ core–shell nanocomposites. Furthermore, the influences of the weight ratio of AN/γ-Al_2_O_3_ on the electrical conductivity, chemical structure, and morphology of the synthesized conductive PANDB/γ-Al_2_O_3_ core–shell nanocomposites are systemically examined using a four-point probe, Fourier transform infrared spectroscopy (FTIR), ultraviolet–visible spectroscopy (UV-Vis), transmission electron microscopy (TEM), and field emission scanning electron microscopy (FE-SEM).

## 2. Experimental

### 2.1. Materials

Aniline (AN), dodecylbenzene sulfonic acid (DBSA), and ammonium persulfate (APS) were purchased from Merck Co., Darmstadt, Germany. The γ-Al_2_O_3_ nanoparticles were purchased from Degussa Co., Frankfurt, Germany. The diameter of an individual γ-Al_2_O_3_ nanoparticle is about 10~30 nm. 

#### 2.1.1. Synthesis of Pure PANDB 

PANDB was directly synthesized by chemical oxidative polymerization according to the modified procedure described by Cao et al. [29] and our previous study [30]. First, 8 g of aniline was mixed with 23 g of DBSA and 400 mL of distilled water to form a uniform milky white dispersion of aniline–DBSA complex in a 1000-milliliter four-neck flat-bottom reactor at room temperature with appropriate stirring. Then, APS solution (20 g of APS dissolved in 200 mL of distilled water) was slowly added to the reactor. The aniline:DBSA:APS molar ratio was 1:0.8:1. After 2 h synthesis, the dark green PANDB dispersion was precipitated by the addition of 600 mL of acetone. The reaction mixture was filtered and washed several times with deionized water until the filtrate was colorless. Finally, the resulting precipitate was collected and dried in an oven at 60 °C for 24 h.

#### 2.1.2. Synthesis of Conductive PANDB/γ-Al_2_O_3_ Core–Shell Nanocomposites

Conductive PANDB/γ-Al_2_O_3_ core–shell nanocomposites were synthesized by in situ chemical oxidative polymerization. Two grams of γ-Al_2_O_3_ nanoparticles were dispersed in 100 mL of an aqueous solution containing 1 g of aniline, and the resulting dispersion was stirred at room temperature for 10 min. The weight ratio of AN/γ-Al_2_O_3_ was 3/1. Then, 50 mL of aqueous DBSA was added to the solution. After stirring for 10 min, 30 mL of APS aqueous solution was added dropwise to the dispersion with constant stirring. The molar ratio of aniline:DBSA:APS was 1:0.8:1. The resulting mixture was reacted at room temperature for 2 h. Subsequently, the product was washed with deionized water until the filtrate became colorless. Afterwards, the product was dried in a vacuum oven at 60 °C for 24 h. To determine the effect of the weight ratio of AN/γ-Al_2_O_3_ on the properties of the PANDB/γ-Al_2_O_3_ core–shell nanocomposites, products with different weight ratios were also applied, namely 3/2, 3/3, 3/4, and 3/5. 

### 2.2. Characterization

#### 2.2.1. Electrical Conductivity Analysis 

A sample of 0.1 g was weighed and then pressed at 3.0 × 10^5^ psi for 2 min at room temperature. Four-point probe measurement (model: LSR4-KHT200, KeithLink Technology Co., Ltd., Taipei, Taiwan) was used to determine the electrical conductivity σ (S/cm) of the sample at room temperature.

#### 2.2.2. FTIR Analysis 

The chemical structure of the samples was detected by Fourier transform infrared spectroscopy (FTIR) (model Spectrum One; Perkin Elmer, Waltham, MA, USA) at 32 scans/s in the wavenumber range of 4000–400 cm^−1^. The powdered sample was ground together with potassium bromide (KBr) (approximately 1:99 by weight) to a fine powder, and the homogeneous mixture was pressed into a pellet for analysis.

#### 2.2.3. UV-Vis Analysis

The synthesized product was dispersed in absolute ethanol with ultrasonic stirring for 1 h at room temperature. A UV-Vis spectrophotometer (UV-Vis; Shimadzu, model UV-2401 PC, Kyoto, Japan) was used to measure the absorption of the sample solution in the wavelength range of 300–900 nm.

#### 2.2.4. TEM Examination

The synthesized PANDB and PANDB/γ-Al_2_O_3_ nanocomposite samples were diluted and dispersed uniformly. They were then cast on carbon-coated copper grids to prepare for transmission electron microscopy (TEM) (JEM-1230, JEOL, Ltd., Tokyo, Japan) analysis. The microscope was operated at an accelerating voltage of 80 kV.

#### 2.2.5. FE-SEM Examination

The samples were coated with a gold-palladium film. The surface morphology of the samples was observed using a field emission scanning electron microscope (JSM 6700F model; JEOL, Ltd., Tokyo, Japan).

## 3. Results and Discussion

Figure 1 clearly shows the color change of the polymerization solution at the weight ratio of AN/γ-Al_2_O_3_ = 3/1 after a reaction time of 3, 27, and 30 min. The color of the initial reaction solution was milky white. After adding the APS solution, the color of the reaction solution remained unchanged for the first 15 min and then turned light blue, blue, and finally turquoise within 15–27 min. This color change indicated the formation of the pernigraniline oxidation state of PANI [31]. After 27 to 30 min, the reaction solution became green to dark green quickly. This showed that the oxidation state of pernigraniline was converted to the oxidation and reduction states of emeraldine of PANI. The color change from green to dark green is due to the doping of DBSA onto the PANI backbone to form conductive PANDB (ES type). 

The color change during the reaction can be used as an indicator of the polymerization rate of aniline. The experimental results show that the time required for color change increased (i.e., the polymerization rate decreased) as the AN/γ-Al_2_O_3_ weight ratio decreased. This observation was due to the fact that the probability of collision between AN and initiator (that is, APS) molecules was decreased while the amount of γ-Al_2_O_3_ was increased in the reaction solution. Therefore, the AN polymerization rate decreased as the weight of γ-Al_2_O_3_ was increased.

Table 1 exhibits the electrical conductivities of pure γ-Al_2_O_3_, PANDB, and the PANDB/γ-Al_2_O_3_ core–shell nanocomposites. Since γ-Al_2_O_3_ has good electrical insulation, its electrical conductivity cannot be detected by four-point probe measurement. Meanwhile, the electrical conductivity of pure PANDB was 0.82 S/cm according to the test results. The results in Table 1 indicate that the electrical conductivity of the PANDB/γ-Al_2_O_3_ core–shell nanocomposites decreased as the weight ratio of AN/γ-Al_2_O_3_ was decreased. When the weight ratio of AN/γ-Al_2_O_3_ was 3/1, the electrical conductivity of the PANDB/γ-Al_2_O_3_ core–shell nanocomposite was about 0.72 S/cm. This result implies that PANDB could be successfully coated onto the surface of γ-Al_2_O_3_ nanoparticles, and the conductivity of γ-Al_2_O_3_ could be improved by forming electrically conductive PANDB/γ-Al_2_O_3_ core–shell nanocomposites. When the weight ratio of AN/γ-Al_2_O_3_ was decreased from 3/1 to 3/5, the conductivity of the PANDB/γ-Al_2_O_3_ core–shell nanocomposite decreased from 0.72 to 0.53 S/cm. This is due to the fact that the thickness of the conductive PANDB on γ-Al_2_O_3_ was relatively decreased with the increasing weight of γ-Al_2_O_3_.

Figure 2 presents photographs of the γ-Al_2_O_3_ nanoparticles (Figure 2a) and the conductive PANDB/γ-Al_2_O_3_ core–shell nanocomposite synthesized at the weight ratio of AN/γ-Al_2_O_3_ = 3/1 through in situ polymerization (Figure 2b). The appearance of the γ-Al_2_O_3_ nanoparticles was white, whereas the appearance of the conductive PANDB/γ-Al_2_O_3_ core–shell nanocomposite was dark green. This is because PANDB is an emeraldine salt (ES) type with good electrical conductivity. Therefore, the color of the conductive PANDB/γ-Al_2_O_3_ core–shell nanocomposite was dark green. 

Figure 3 shows the FTIR spectra of pure PANDB and the conductive PANDB/γ-Al_2_O_3_ core–shell nanocomposites synthesized at the weight ratios of AN/γ-Al_2_O_3_ = 3/2 and 3/5. Peaks at about 2853 and 2953 cm^−1^ were observed due to the stretching vibration mode of the -CH (-CH_3_ or -CH_2_-) for all samples. For pure PANDB (Figure 3a), the characteristic peaks at 1567 and 1491 cm^−1^ were due to the stretching vibrations of the N=Q=N and the N-B-N ring, respectively. The characteristic peak at 1300 cm^−1^ was attributed to the C-N stretching vibrations of the secondary amine in the main chain of PANDB. The peaks at 998–1040 cm^−1^ were due to the asymmetric and symmetric O=S=O stretching vibrations of DBSA. Characteristic peaks at 1100–1200 cm^−1^ were due to the B-NH-Q bond or the B-NH-B bond and the in-plane bending vibration of benzenoid or quinonoid C-H bonds (where B represents benzenic-type rings and Q represents quinonic-type rings). As these absorption peaks overlapped in the range of 1000 to 1200 cm^−1^, a broad peak was observed. The peaks at 800–700 cm^−1^ were attributed to the characteristic feature of the B-NH-Q bond or the B-NH-B bond and the out-of-plane bending vibration of benzenoid or quinonoid -CH and –N–H bonds. These characteristic peaks are in good agreement with those reported in the literature [32,33]. The bands in the range of 400~600 cm^−1^ were associated with γ-Al_2_O_3_ nanoparticles. Note that there was no significant interaction between PANDB molecules and γ-Al_2_O_3_ nanoparticles from the FTIR results. The good adhesion of PANDB to γ-Al_2_O_3_ nanoparticles was assumed to be physicochemical in nature [21].

Figure 4 shows the UV-Vis absorption spectra of pure PANDB and the PANDB/γ-Al_2_O_3_ core–shell nanocomposites. For pure PANDB, three characteristic absorption peaks can be clearly observed in the UV-Vis spectrum at ~340, ~430, and ~830 nm (Figure 4a). The absorption peak at ~340 nm was due to the *π*–*π** transition of the benzenoid rings, while the peaks at ~430 and ~830 nm were attributed to the polaron–*π** transition and *π*–polaron transition, respectively [34,35]. The analytical results show that the synthesized PANDB was an emeraldine salt (ES) form. Three characteristic absorption peaks also appeared in the spectrum of PANDB/γ-Al_2_O_3_ core–shell nanocomposites owing to the presence of PANDB (Figure 4b–f). However, the absorption peak related to the *π*–polaron transition was shifted to a lower wavelength (from 840 to 820 nm) by decreasing the weight ratio of AN/γ-Al_2_O_3_ from 3/1 to 3/5_._ This result was attributed to less PANDB being coated on the surface of the γ-Al_2_O_3_ nanoparticles when the amount of γ-Al_2_O_3_ was increased. This result is consistent with the decrease in conductivity.

Figure 5 and Figure 6 show the TEM and FE-SEM images of pure PANDB (Figure 5a and Figure 6a), γ-Al_2_O_3_ nanoparticles (Figure 5b and Figure 6b), and the PANDB/γ-Al_2_O_3_ core–shell nanocomposites synthesized at the weight ratios of AN/γ-Al_2_O_3_ = 3/1 and 3/3 (Figure 5c,d and Figure 6c,d). Figure 5a and Figure 6a demonstrate that pure PANDB was aggregated by irregular particles and rod-like PANDB. Figure 5b shows an image of independent γ-Al_2_O_3_ nanoparticles and aggregates of γ-Al_2_O_3_ nanoparticles with irregular shapes due to the high surface energy of the nanoparticles. Figure 5b and Figure 6b show that the diameter of individual γ-Al_2_O_3_ nanoparticles was about 10~30 nm and the diameter of the clusters of γ-Al_2_O_3_ nanoparticles ranged from 30 to 100 nm. Comparing Figure 5c with Figure 5d or Figure 6c with Figure 6d, the surface of the γ-Al_2_O_3_ nanoparticle was coated with less PANDB as the amount of γ-Al_2_O_3_ nanoparticle was increased. Therefore, the electrical conductivity of the PANDB/γ-Al_2_O_3_ core–shell nanocomposites decreased as the amount of γ-Al_2_O_3_ nanoparticle was increased. 

Based on the result of the TEM images, the synthesis of conductive PANDB/γ-Al_2_O_3_ core–shell nanocomposites via an in situ polymerization process is shown in Figure 7. First, the TEM results indicated that the γ-Al_2_O_3_ nanoparticles were coated with an anilinium–DBSA complex. Then, by adding APS for polymerization, conductive PANDB/γ-Al_2_O_3_ core–shell nanocomposites could be obtained. Furthermore, the independent γ-Al_2_O_3_ nanoparticles and the aggregates of γ-Al_2_O_3_ nanoparticles were simultaneously coated by PANDB to form conductive PANDB/γ-Al_2_O_3_ core–shell nanocomposites.

## 4. Conclusions

In this study, an in situ polymerization method was used to prepare electrically conductive PANDB/γ-Al_2_O_3_ core–shell nanocomposites. The core is γ-Al_2_O_3_, and the shell is PANDB. The electrical conductivity of pure PANDB was 0.82 S/cm. The electrical conductivity of the PANDB/γ-Al_2_O_3_ core–shell nanocomposite decreased with increasing the amount of γ-Al_2_O_3_. When the weight ratio of AN/γ-Al_2_O_3_ was decreased from 3/1 to 3/5, the conductivity of the PANDB/γ-Al_2_O_3_ core–shell nanocomposite decreased from 0.72 to 0.53 S/cm. Both the TEM and FE-SEM images showed that PANDB could be successfully coated on the surface of γ-Al_2_O_3_ nanoparticles. Therefore, the electrical conductivity of γ-Al_2_O_3_ could be improved by forming a conductive PANDB/γ-Al_2_O_3_ core–shell nanocomposite.

## Figures and Tables

**Figure 1 polymers-14-02232-f001:**
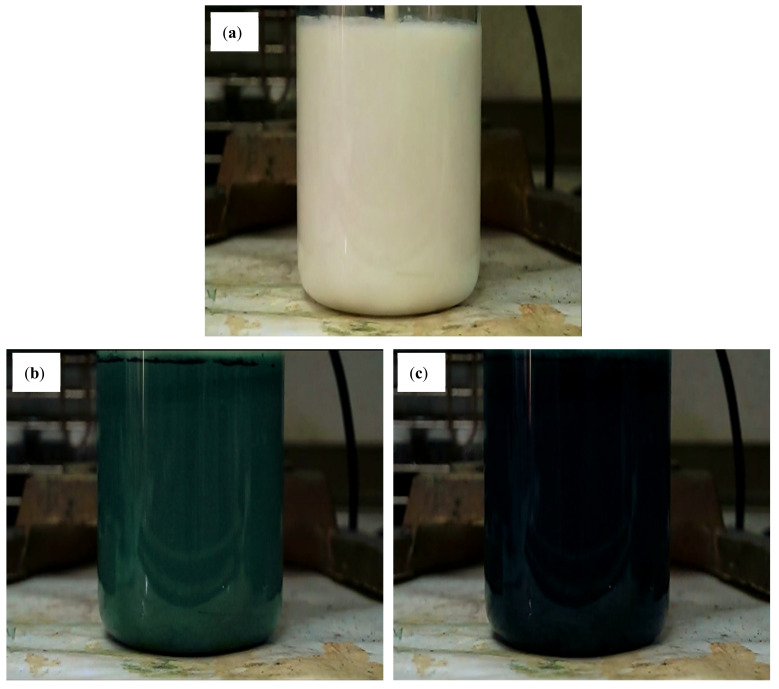
Color changes of the polymerization solution at the weight ratio of AN/γ-Al_2_O_3_ of 3/1 after (**a**) 3, (**b**) 27, and (**c**) 30 min reaction time.

**Figure 2 polymers-14-02232-f002:**
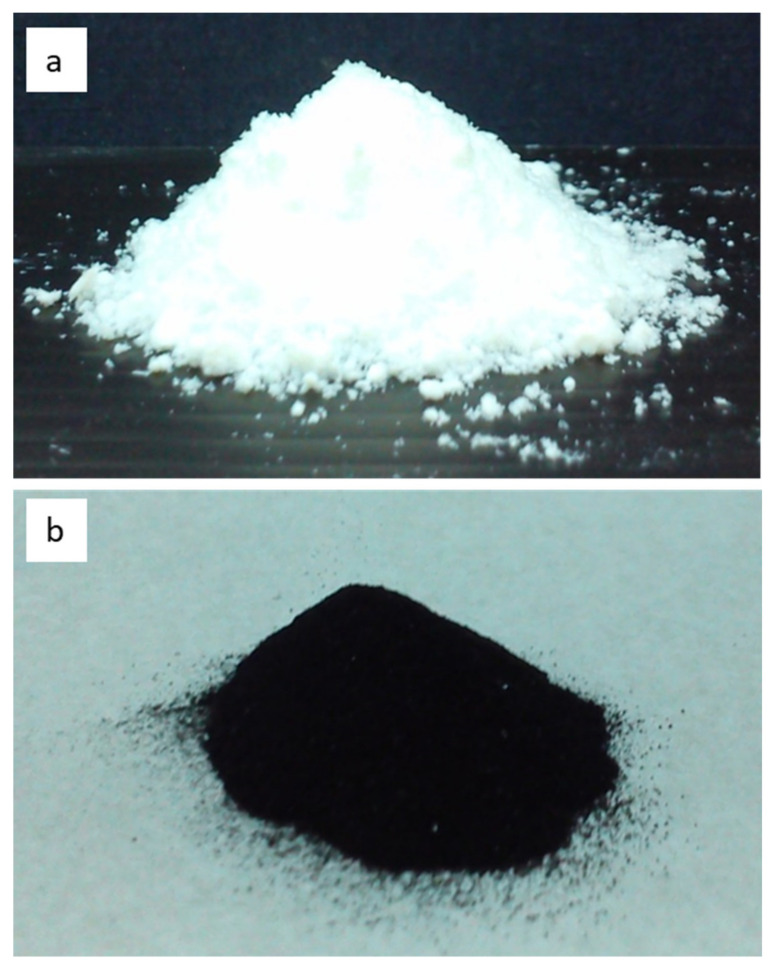
Photographs of (**a**) γ-Al_2_O_3_ nanoparticles and (**b**) conductive PANDB/γ-Al_2_O_3_ core–shell nanocomposite synthesized via in situ polymerization (weight ratio of AN/γ-Al_2_O_3_ = 3/1).

**Figure 3 polymers-14-02232-f003:**
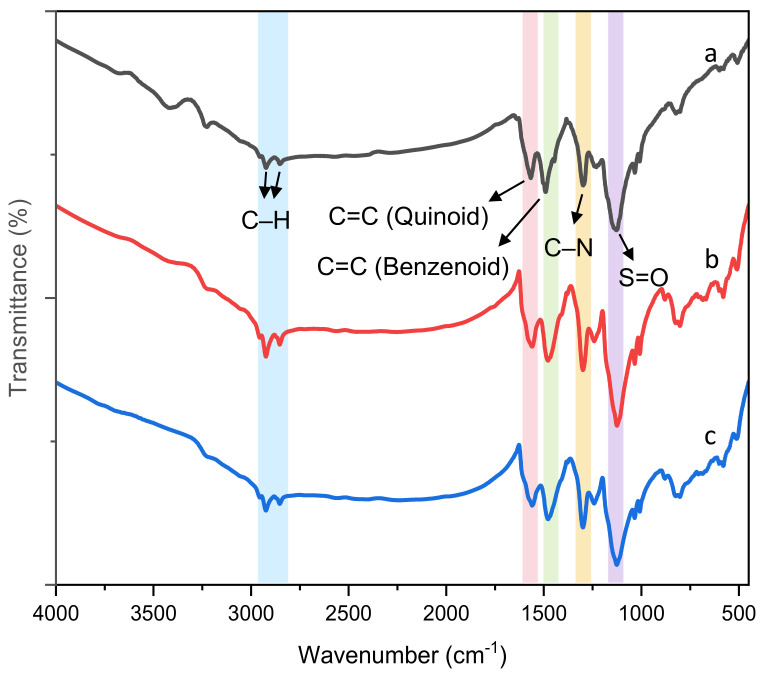
FTIR spectra of pure PANDB (**a**) and conductive PANDB/γ-Al_2_O_3_ core-shell nanocomposites synthesized at the weight ratios of AN/γ-Al_2_O_3_ = (**b**) 3/2 and (**c**) 3/5.

**Figure 4 polymers-14-02232-f004:**
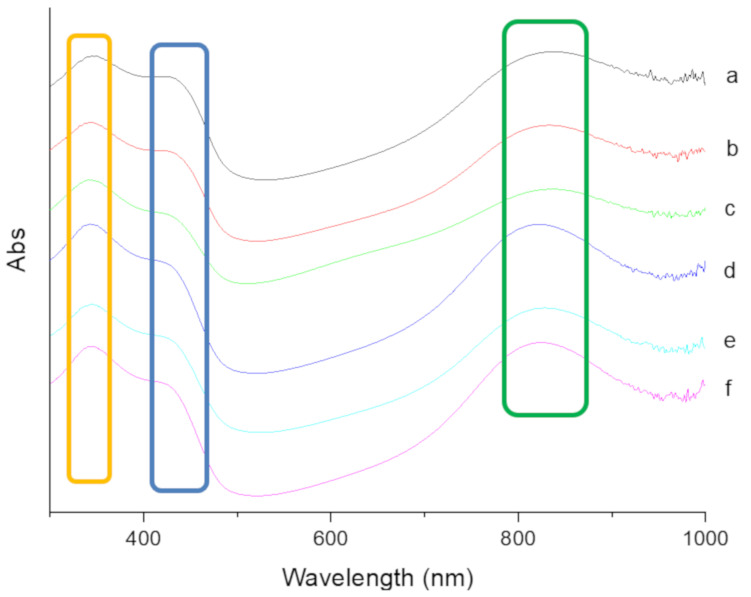
UV-Vis spectra of (**a**) pure PANDB and conductive PANDB/γ-Al_2_O_3_ core–shell nanocomposites synthesized at the weight ratios of AN/γ-Al_2_O_3_ = (**b**) 3/1, (**c**) 3/2, (**d**) 3/3, (**e**) 3/4, and (**f**) 3/5.

**Figure 5 polymers-14-02232-f005:**
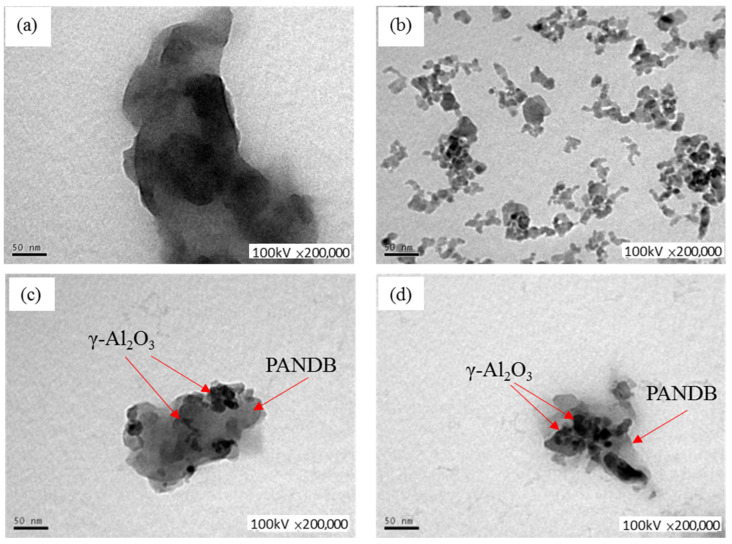
TEM images of (**a**) pure PANDB, (**b**) γ-Al_2_O_3_, and conductive PANDB/γ-Al_2_O_3_ core–shell nanocomposites synthesized at weight ratios of AN/γ-Al_2_O_3_ = (**c**) 3/1 and (**d**) 3/3.

**Figure 6 polymers-14-02232-f006:**
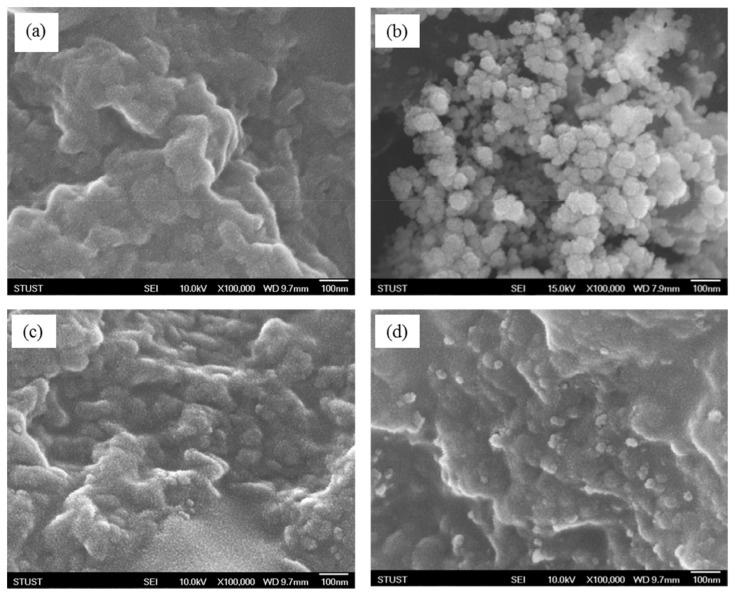
FE-SEM images of (**a**) pure PANDB, (**b**) γ-Al_2_O_3_, and conductive PANDB/γ-Al_2_O_3_ core–shell nanocomposites synthesized at weight ratios of AN/γ-Al_2_O_3_ = (**c**) 3/1 and (**d**) 3/3.

**Figure 7 polymers-14-02232-f007:**
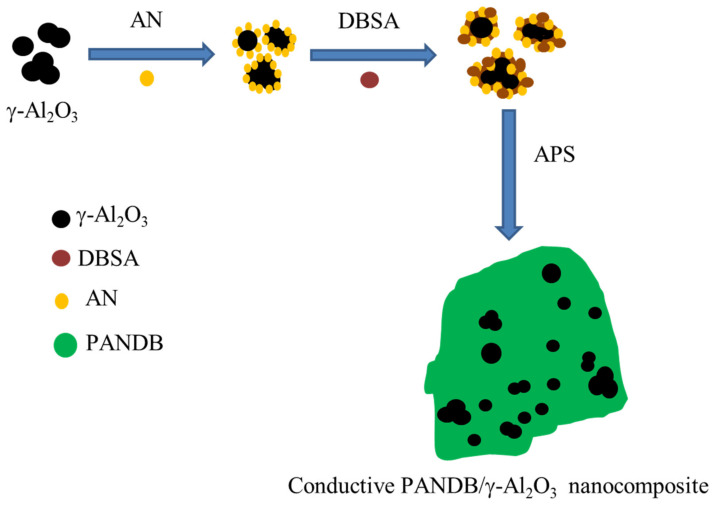
Synthesis process of conductive PANDB/γ-Al_2_O_3_ core–shell nanocomposites through in situ polymerization.

**Table 1 polymers-14-02232-t001:** Influence of the weight ratio of AN/γ-Al_2_O_3_ on the electrical conductivity of PANDB/γ-Al_2_O_3_ core–shell nanocomposites.

Weight Ratio of AN/γ-Al_2_O_3_	Conductivity of PANDB and PANDB/γ-Al_2_O_3_ Nanocomposites (S/cm)
Pure γ-Al_2_O_3_	-
Pure PANDB	0.82
3/1	0.72
3/2	0.58
3/3	0.57
3/4	0.55
3/5	0.53

## Data Availability

Not applicable.

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
