# Peer review of "Surface Modification of γ-Al2O3 Nanoparticles Using Conductive Polyaniline Doped by Dodecylbenzene Sulfonic Acid"

_polymers, 2022, doi:10.3390/polym14112232_

Round 1

Reviewer 1 Report

The topic of the manuscript is very interesting and also obtained results are bringing the advancement into the studied subject. However, there are few issues which decrease the overall quality of the manuscript and they have to be corrected.

  1. There are introduced and explained abbreviation within the Abstract. The abbreviations should be eliminated. By the way, all of them are introduced and explained alo in an Introduction part.
  2. Section 2.1: supplier, CITY and country of origin has to be mentioned for each used reagent.
  3. section 2.2: for all instruments, the type, manufacturer, CITY and country of origin has to be mentioned. 

All in all, the manuscript is interesting and could be accepted for publication after above mentioned corrections.

Author Response

Many thanks for the comments of reviewers. We had made corrections and responses to the comments listed as below. All corrections are showed as red color.

The topic of the manuscript is very interesting and also obtained results are bringing the advancement into the studied subject. However, there are few issues which decrease the overall quality of the manuscript and they have to be corrected.

1. There are introduced and explained abbreviation within the Abstract. The abbreviations should be eliminated. By the way, all of them are introduced and explained also in an Introduction part.

Response: It had been corrected.

2. Section 2.1: supplier, CITY and country of origin has to be mentioned for each used reagent.

Response: It had been mentioned.

3. section 2.2: for all instruments, the type, manufacturer, CITY and country of origin has to be mentioned. 

Response: It had been mentioned.

All in all, the manuscript is interesting and could be accepted for publication after above mentioned corrections.

Reviewer 2 Report

The manuscript reports Surface modification of gamma Alumina nanoparticles using conductive Polyaniline doped by dodecylbencen sulfonic acid, it sound an interesting work but there are several questions that need to be improved before the manuscript can continue the process for publication, following they are detailed:

- First time an abbreviation is written must be defined, for instance SEM, TEM,  FE-SEM, among others, please check it.

- I recommend to check some of the next references, it would be useful for discussion of results and as state of art of use of nanoparticles with PANI.

https://doi.org/10.1016/j.matpr.2021.03.182

https://doi.org/10.1002/app.24326

https://doi.org/10.1016/j.synthmet.2018.01.006

https://doi.org/10.1063/5.0060886

-In line 111, please correct the range for FTIR analysis, the correct use be range from 4000-400 cm-1.

- It would be recommended to carry out Raman spectroscopy characterization in the aim to corroborate the formation of Emeraldine salt of PANI, due FTIR results are not enough conclusive, due the spectra plotted have not good quality, the baseline need to be corrected and it would be recommended to use arrows to indicate the peaks that show changes. Usually Raman is more useful technique for characterization of PANI.

- The UV vis spectra shows that intensity of peak at 820 nm attributed to π polaron has not a good tendency  is not in the way is plotted, for instance sample c (ratio 3/2) has a low intensity ad sample f peak has almost the same intensity than sample a and b, please correct this or justify this behavior. Usually the displacement to lower wavelength is result of electronic transition that can be associated to conductive properties.

Author Response

Many thanks for the comments of reviewers. We had made corrections and responses to the comments listed as below. All corrections are showed as red color.

The manuscript reports Surface modification of gamma Alumina nanoparticles using conductive Polyaniline doped by dodecylbenzene sulfonic acid, it sounds an interesting work but there are several questions that need to be improved before the manuscript can continue the process for publication, following they are detailed:

- First time an abbreviation is written must be defined, for instance SEM, TEM, FE-SEM, among others, please check it.

Response: It had been checked and defined.

- I recommend to check some of the next references, it would be useful for discussion of results and as state of art of use of nanoparticles with PANI.

https://doi.org/10.1016/j.matpr.2021.03.182

https://doi.org/10.1002/app.24326

https://doi.org/10.1016/j.synthmet.2018.01.006

https://doi.org/10.1063/5.0060886

Response: Many thanks for the comment. The first three articles have been cited. The final one cannot be found.

-In line 111, please correct the range for FTIR analysis, the correct use be range from 4000-400 cm-1.

Response: It had been corrected.

- It would be recommended to carry out Raman spectroscopy characterization in the aim to corroborate the formation of Emeraldine salt of PANI, due FTIR results are not enough conclusive, due the spectra plotted have not good quality, the baseline need to be corrected and it would be recommended to use arrows to indicate the peaks that show changes. Usually Raman is more useful technique for characterization of PANI.

Response: Figure 3 had been replotted and explained in more detail.

- The UV vis spectra shows that intensity of peak at 820 nm attributed to π polaron has not a good tendency is not in the way is plotted, for instance sample c (ratio 3/2) has a low intensity and sample f peak has almost the same intensity than sample a and b, please correct this or justify this behavior. Usually the displacement to lower wavelength is result of electronic transition that can be associated to conductive properties.

Response: Many thanks for the comments. The intensity of peak at about 820 nm is not the only way to analyze the conductivity of PANI. In general, the higher the peak shift to higher wavelengths (red shift), the higher the conductivity. Another analysis method is the Q/B ratio. The intensity ratio of Q peak (~820 nm) to B peak (~340 nm) is called Q/B ratio, and proportional to the ratio of quinoid units to benzenoid units in PANI. The higher Q/B ratio, the higher the conductivity.

In this research, although the electrical conductivity of PANDB/γ-Al2O3 nanocomposites decreased with the increase of Al2O3 content, the difference was small. Therefore, the positions of the characteristic peaks in the UV-Vis spectra of all products were similar. In the text, we have illustrated “the absorption peak related to π–polaron transition was shifted to the lower wavelength (from 840 to 820 nm) by decreasing the weight ratio of AN/γ-Al2O3 from 3/1 to 3/5”.

Round 2

Reviewer 2 Report

After review the corrected version, this shows a significant improvement and recommendations/observations were taken in account, so I recommend to Accept the manuscript as it.